# STREAM-NATIVE MACHINE UNLEARNING

## ABSTRACT

Machine unlearning work assumes a static, i.i.d. training environment that doesn't truly exist. Machines are expected to learn and unlearn We introduce the *Memory Pair*, the first stream-native unlearning algorithm coupling an online learning-unlearning pair. Our framework achieves state-of-the-art logarithmic regret bounds under strong convexity that scale naturally for nonstationary stream environments. These guarantees extend the lifespan of deployed models, reducing costly retraining while ensuring compliance with data deletion requirements. Finally, the algorithm intentionally achieves a novel sublinear memory footprint for an online machine unlearning. This makes it attractive for on-device learning, memory-constrained devices, and IoT networks where privacy is paramount.

## 1 INTRODUCTION

Modern machine learning systems increasingly operate in streaming and distributed environments. Rather than train once on a fixed dataset, models are continuously updated from event-driven pipelines, edge devices, and federated systems. This reality conflicts with the assumptions of most machine unlearning algorithms, which are typically designed for static, batch settings (Bourtoule et al., 2019; Ginart et al., 2019; Neel et al., 2020; Sekhari et al., 2021; Guo et al., 2020). At the same time, regulatory pressures (e.g., the GDPR "right to be forgotten") motivate mechanisms that make a model behave as if deleted data had never been observed, with formal indistinguishability guarantees often framed through differential privacy and related tools (Dwork & Roth, 2014; Bun & Steinke, 2016; Chen et al., 2020).

The challenge is clear: existing unlearning techniques either retrain from scratch or approximate batch removal with offline corrections (Bourtoule et al., 2019; Ginart et al., 2019; Guo et al., 2020; Neel et al., 2020; Sekhari et al., 2021). Both approaches struggle in settings where learning and unlearning must be interleaved with live prediction, and they can incur nontrivial accuracy or latency costs. We address this gap with the **Memory Pair**, a unified stream-native learning and unlearning framework. By coupling a learner and an unlearner around a shared state, the Memory Pair provides a regret-optimal treatment of machine unlearning in dynamic data streams (log-static, linear-path) while remaining compatible with certification mechanisms in the style of DP/zCDP (Dwork & Roth, 2014; Bun & Steinke, 2016).

This is the natural next step for a body of research that sees learning and unlearning as deeply dependent functions. Modern unlearning algorithms explicitly use the gradients from learning to maintain privacy. The difference with the Memory Pair is that it doesn't store an entire stream's worth of data, while still utilizing efficiently utilizing second-order information.

Our contributions are as follows:

- We introduce the **Memory Pair framework**, which achieves *logarithmic regret* under $\lambda$-strong convexity, even with certified deletions interleaved throughout the stream, drawing on online OCO techniques and quasi-Newton preconditioning (Cesa-Bianchi & Lugosi, 2006; Liu & Nocedal, 1989; Mokhtari & Ribeiro, 2014; Schraudolph et al., 2007; Byrd et al., 2015; Moritz et al., 2016).

- We propose a **deletion capacity odometer**, an adaptive accounting mechanism that tracks the interplay between learning progress and fidelity to retraining, aligning with certification norms (Dwork & Roth, 2014; Bun & Steinke, 2016).

- We demonstrate a **constant-memory implementation** via online L-BFGS, suitable for streaming and edge-device deployment (Liu & Nocedal, 1989; Mokhtari & Ribeiro, 2014; Byrd et al., 2015; Moritz et al., 2016).

- We provide $(\varepsilon, \delta)$**- or zCDP-style indistinguishability guarantees**, ensuring that each unlearning update is statistically equivalent to retraining from scratch (Dwork & Roth, 2014; Bun & Steinke, 2016).

Together, these results establish the Memory Pair as an online algorithm that balances accuracy, memory efficiency, and certification-friendly guarantees for continuous machine unlearning (Qiao et al., 2024; Zhang et al., 2025; Waerebeke et al., 2025).

## 2 MEMORY PAIR ALGORITHM

**Definition 1** (Memory Pair). *Let* $\{E_t\}_{t=1}^N$ *be an event stream where* $E_t \in \{insert(x_t, y_t), delete(u_t)\}$. *A pair of algorithms* $(A, \bar{A})$ *with shared state* $\theta_{t-1}$ *acts as*

$$\text{learn step: } \theta_t = A(\theta_{t-1}, E_t), \qquad \text{unlearn step: } \bar{\theta}_t = \bar{A}(\theta_{t-1}, E_t).$$

*Denote by* $\tilde{\theta}_t$ *the ideal replay model, i.e. the ideal model retrained from scratch without the offending data. Fix an regret target* $\gamma$, *confidence* $\delta$, *fidelity budget* $(\varepsilon^*, \delta^*)$, *and a deletion capacity* $m$. *Then* $(A, \bar{A})$ *is a* $(\gamma, m, \varepsilon^*, \delta^*)$*-memory pair if, for every event stream and for every horizon* $N$, *the following hold:*

1. *Fidelity. The strength of the guarantees depends entirely on the context. Stronger guarantees will require higher sample complexities and lower deletion capacities.*

$$\Pr\left[\tfrac{1}{N}R_N(\theta) \le \gamma\right] \ge 1 - \delta.$$

2. $(\varepsilon, \delta)$*-Certified Unlearning. Let* $\mathcal{M}_{del}$ *denote the (randomized) model state produced after applying a delete operation, and let* $\mathcal{M}_{retrain}$ *denote the model state produced by retraining from scratch on the dataset with the deleted point removed. We say the algorithm achieves* $(\varepsilon, \delta)$*-certified unlearning if for every measurable set* $S$,

$$\Pr[\mathcal{M}_{del} \in S] \le e^\varepsilon \Pr[\mathcal{M}_{retrain} \in S] + \delta.$$

*By post-processing invariance, the same guarantee holds for any function of the model state (e.g., predictions).*

3. *m-deletion capacity. Let* $D_N$ *be the number of* DELETE *events up to* $N$. *If* $D_N \le m$, *then Conditions 1 and 2 hold.*

The algorithm combines the regret guarantees of online learning with the DP-style language that are ubiquitous in machine unlearning. Bundling the learning and unlearning algorithms is a natural extension to a body of work that has increasingly found them inseparable. Indeed, recent Hessian-free methods of machine unlearning explicitly use the learning process to gain gradient information that is later used to unlearn Qiao et al. (2024).

### 2.1 SYMMETRY OF INSERT AND DELETE

A key design principle is that insert and delete operations share an identical control flow: both compute a gradient, obtain a quasi-Newton direction, and update the model parameters using the same curvature memory. The only differences are (i) the sign of the update, and (ii) whether calibrated noise and odometer accounting are applied. This symmetry eliminates the need to maintain duplicate state and ensures that the information used to learn a point is precisely the information later needed to unlearn it. From an implementation perspective, it means the delete operator is not an afterthought but a first-class mirror of the insert operator.

---

**Algorithm 1** Memory Pair Step (Insert/Delete)

---

**Input:** $(x, y)$, model state $(\theta, \text{lbfgs})$, step size $\alpha$
Compute gradient $g \leftarrow \nabla_\theta \ell(\theta; x, y)$
Compute direction $d \leftarrow \text{lbfgs.direction}(g)$
**if** Insert **then**
    $\theta \leftarrow \theta + \alpha d$
    Update curvature pairs $(v, r)$
**else if** Delete **then**
    $\theta \leftarrow \theta - \alpha d + \eta$, where $\eta \sim \mathcal{N}(0, \sigma^2 I)$
    Update curvature pairs (optional downdate)
    Decrement deletion capacity via odometer
**end if**

---

## 2.2 Deletion Capacity Accounting via an Odometer

A critical component of the Memory Pair framework is its ability to manage the trade-off between unlearning fidelity and model regret. This is accomplished through a strict accounting mechanism that we term a **deletion capacity odometer**. We model deletion-capacity as a dynamic amount that changes based on the stream. Unlike static analyses, we don't envision that a model has a fixed deletion capacity after a static training period. This odometer regulates the model's utility (i.e., its regret bounds) in response to deletions. It is important to note that $(\varepsilon, \delta)$-certified unlearning budgets are monotone and do not replenish; our "replenishment" language refers strictly to utility. As additional data insertions improve regret bounds (e.g., by increasing $N$), the system can tolerate more deletions before violating its target utility threshold.

By first proving the most conservative case of regret bounds, we then show that strong convexity and dynamic regret analyses allow for $\mathcal{O}(\sqrt{T})$ bounds on deletion capacity and sample complexity. For the sake of simplicity, we use the Zero-Concentrated Differential Privacy definition to scale the amount of noise used for deletions to the influence of the unlearned point.

## 2.3 Composition under zero-Concentrated DP

**Capacity as a $\rho$-budget.** We use Zero-Concentrated Differential Privacy (zCDP) Bun & Steinke (2016) for internal accounting due to its tighter composition properties, then convert to the standard $(\varepsilon, \delta)$-certified guarantee for reporting. Fix a total fidelity budget $\rho_{\text{tot}} > 0$. The model is initialized with $m$ deletion capacity and we allocate the budget *uniformly*,

$$\rho_{\text{s}} = \frac{\rho_{\text{tot}}}{m},$$

so that after at most $m$ deletions the cumulative privacy loss is $\sum_{j=1}^{m} \rho_{\text{s}} = \rho_{\text{tot}}$ by the additive composition rule of zCDP.

**Per-delete noise calibration.** In each DELETE$(u)$ the algorithm adds Gaussian noise $\eta \sim \mathcal{N}\left(0, \sigma_{\text{s}}^2 \mathbf{I}_d\right)$ with scale

$$\sigma_{\text{s}} = \frac{S_{\text{step}}}{\sqrt{2\rho_{\text{s}}}},$$

where $S_{\text{step}}$ is a global $\ell_2$-sensitivity bound on the unlearning update derived from the loss function's properties (e.g., $S_{\text{step}} \le G/\lambda$ under strong convexity). The runtime accountant tracks deletions and the spent fidelity

$$\rho_{\text{spent}} = \texttt{deletions\_so\_far} \times \rho_{\text{s}}.$$

A new deletion is rejected with a RuntimeError once

$$\rho_{\text{spent}} + \rho_{\text{s}} > \rho_{\text{tot}}.$$

**Why a finite $m$ is necessary.** Each deletion increases both (i) the cumulative zCDP loss and (ii) the *variance* of the model parameters through $\sigma_{\text{s}}^2$. Beyond a problem-dependent threshold, the injected noise dominates the learning signal and jeopardizes the algorithm's average-regret bound $\gamma$ (**??**). The $m$-deletion capacity therefore matches the largest $m$ for which Theorems 3 and 4 remain valid.

## 2.4 Design Intuition

Treating learning and unlearning as separate modules typically leads to duplicated state and inconsistent updates. By contrast, the Memory Pair maintains a shared representation of curvature information, ensuring that learning and unlearning reinforce one another. The result is near-instantaneous deletions, logarithmic regret growth under strong convexity, and a principled capacity mechanism that certifies when retraining is unavoidable.

## 3 Theoretical Results

Our analysis shows that the Memory Pair preserves hallmark guarantees of online convex optimization—logarithmic *static* regret under strong convexity—while incorporating new terms that capture the effect of deletions and nonstationary data. We sharpen the bounds by decomposing regret into two pieces: a static term due to learning dynamics and a *pathwise* term due to distributional drift. We further introduce adaptive capacity bounds that quantify how long a model can sustain reliable unlearning before retraining becomes necessary.

The key difference between general convexity and $\lambda$-strong convexity is the quadratic growth condition, which provides stability of the gradient and enables a decreasing learning-rate schedule. We choose $\eta_t = (\lambda t)^{-1}$, which yields a telescoping potential plus a harmonic sum and hence a logarithmic bound on the static component. Full proofs appear in the Appendix and rely on the bounded-spectrum preconditioner from online L-BFGS (Assumptions 1 and 2).

**Theorem 1** (Logarithmic cumulative regret with $m$ certified deletions). *Let $\{\ell_t\}_{t=1}^T$ be $\lambda$-strongly convex and $G$-Lipschitz over a closed, convex domain $\mathcal{W}$ of diameter $D$. Assume the inverse-Hessian preconditioners maintained by online L-BFGS satisfy uniform eigenvalue bounds $cI \preceq B_t^{-1} \preceq CI$ (see Assumption 2). Use the step size $\eta_t = \frac{1}{\lambda t}$. Suppose at most $m$ deletions occur at arbitrary times, and each deletion is implemented by adding Gaussian noise with scale $\sigma_{step}$, calibrated to a target $(\varepsilon^*, \delta^*)$ via standard composition. Then for any static comparator $w^* \in \arg\min_{w \in \mathcal{W}} \sum_{t=1}^T \ell_t(w)$, with probability at least $1 - \delta^* - \delta_B$,*

$$R_T(m) := \sum_{t=1}^T \big(\ell_t(w_t) - \ell_t(w^*)\big) \leq \frac{G^2}{\lambda c}\big(1 + \ln T\big) + \Delta_m, \qquad \Delta_m = m\,G\,\sigma_{step}\,\sqrt{2\ln(1/\delta_B)}.$$

*In particular, $R_T(m) = \mathcal{O}\big(\frac{G^2}{\lambda}\ln T + m\,G\,\sigma_{step}\sqrt{\ln(1/\delta_B)}\big)$.*

*Discussion.* The first term is the familiar $\mathcal{O}(\frac{G^2}{\lambda}\ln T)$ rate for strongly convex OCO. The second term $\Delta_m$ is the cumulative effect of $m$ certified deletions, each realized as a noisy quasi-Newton "negative step," scaling linearly in $m$ and with the noise variance.

**Theorem 2** (Dynamic regret under $\lambda$-strong convexity). *Let Assumptions 1 and 2 hold and let $\{\ell_t\}_{t=1}^T$ be $\lambda$-strongly convex and $G$-Lipschitz. Run Algorithm 1 with $\eta_t = (\lambda t)^{-1}$. Then for any comparator path $\{w_t^*\}_{t=1}^T$,*

$$R_T^{\mathrm{dyn}} \leq \underbrace{\frac{G^2}{\lambda c}\big(1 + \ln T\big)}_{\text{static term}} + \underbrace{G\,P_T}_{\text{pathwise term}}, \;\text{ and } R_T^{\mathrm{dyn}} = \mathcal{O}\big(\frac{G^2}{\lambda}\ln T + GP_T\big).$$

Dynamic regret measures the gap to a drifting oracle $\{w_t^*\}$. The static term matches Theorem 1; the pathwise term grows with the realized drift $P_T$. When $P_T = 0$ (stationary data), the bound reduces to Theorem 1. Under gradual drift ($P_T = o(T)$), the algorithm remains competitive; for adversarial drift ($P_T = \Omega(T)$) the pathwise term dominates.

**Static and Dynamic Regret Decomposition.** A key piece of stream-native learning involves building for nonstationary environments. The decomposition above does just that. What appears to be a single cumulative regret curve is a function of the stream's changing conditions. **Our experiments later mimic the nonstationarity of production streams and find consistent convergence of average regret.**

**Theorem 3** ($\gamma$-**Deletion Capacity Bound**). *Fix a target average regret $\gamma > 0$, confidence $\delta_B \in (0, 1)$, and stream length $N$. Let $S_N = \sum_{t=1}^{N} \|g_t\|^2$. Then any $m$ satisfying*

$$m \leq \frac{\gamma N - GD\sqrt{cC\,S_N}}{G\,\sigma_{step}\sqrt{2\ln(1/\delta_B)}} \tag{1}$$

*guarantees $\frac{1}{N}R_N(m) \leq \gamma$ with probability at least $1 - \delta_B$.*

This formalizes the *odometer*: each deletion spends utility via its noise contribution. The admissible $m$ depends on the horizon $N$ and gradient geometry through $S_N$.

**Theorem 4** ($\gamma$-**Sample Complexity Bound**). *Fix $\gamma > 0$, a maximum of $m \in \mathbb{N}$ deletions, and confidence $\delta_B \in (0, 1)$. Let $S_N = \sum_{t=1}^{N} \|g_t\|^2$ and suppose (i) the adaptive regret bound and (ii) the aggregate deletion-noise contribution hold with probability at least $1 - \delta_B$. Then any horizon $N$ such that*

$$\frac{1}{N}R_N(m) = \frac{GD\sqrt{cC\,S_N} + \Delta_m}{N} \leq \gamma \tag{2}$$

*guarantees $\frac{1}{N}R_N(m) \leq \gamma$ with probability $\geq 1 - \delta_B$.*

This flips the viewpoint: for a target $\gamma$ and tolerance $m$, how much data is needed? Via $S_N$, the requirement shrinks on "easy" streams (decaying gradients); in the worst case ($S_N = G^2 N$) it matches classical $\tilde{\mathcal{O}}(\sqrt{N})$ behavior.

**Continuity.** The proofs of Theorems 1–4 follow the preconditioned one-step inequality, telescoping potentials under $\eta_t = (\lambda t)^{-1}$, and zCDP calibration for deletes; see the Appendix (Assumptions 1, 2 and Lemma 2).

**Summary.** Together, Theorems **??**–4 establish that the Memory Pair:

- preserves logarithmic regret despite deletions,
- adapts to drift via a pathwise comparator,
- provides an explicit deletion capacity budget, and
- admits adaptive sample-complexity bounds that exploit benign data.

## 4 EXPERIMENTAL ANALYSIS

Our experiments validate the theoretical decomposition of regret into a logarithmic *static* term and a *pathwise* term that scales with drift. We construct nonstationary streams to mimic production settings, and we instrument the learner with standardized analyses (theory tracking, stepsize validation, privacy/odometer checks, and seed-stability audits). Across all settings, the learner satisfies both regret and privacy guarantees while serving mixed insert/delete workloads.

### 4.1 SETUP

**Data stream with controlled drift.** Rather than fixed datasets (e.g., MNIST), we generate a reproducible stream where the per-step loss $\ell_t(\cdot)$ derives from a distribution that drifts along a parametrized path. The drift is governed by a target path length $P_T$ and realized path length $P_T^{\text{true}} = \sum_{t=2}^{T} \|w_t^\star - w_{t-1}^\star\|_2$. We vary:

1. **Drift regime:** piecewise-constant, linear, and bursty (shock) drift with matched $P_T^{\text{true}}$;
2. **Curvature:** $\lambda \in \{10^{-4}, 10^{-3}, 10^{-2}\}$ to probe strong-convexity strength;
3. **Noise/privacy:** per-event Gaussian noise calibrated by a $\rho$-zCDP odometer, with total budget $\rho_{\text{tot}} \in \{0.1, 0.5, 1.0\}$;
4. **Workload:** interleavings of inserts and deletions at deletion rate $m/T \in \{0, 0.05, 0.1\}$.

**Comparators and decomposition.** We report (i) cumulative regret $R_T = \sum_{t=1}^{T} \left( \ell_t(w_t) - \ell_t(u_t) \right)$ against a *dynamic* comparator $u_t = w_t^\star$ with path length $P_T$, and (ii) the decomposition

$$R_T = R_T^{\text{stat}} + R_T^{\text{path}}, \qquad R_T^{\text{path}} \approx \hat{G} \, P_T^{\text{true}}, \tag{3}$$

where $\hat{G}$ is an empirical Lipschitz proxy (Section 4.1). For strongly convex losses, theory predicts

$$R_T^{\text{stat}} = O\left( \tfrac{G^2}{\lambda} \ln T \right), \qquad \frac{1}{T} R_T^{\text{stat}} \to 0, \tag{4}$$

and a total dynamic regret $R_T = O\left( \tfrac{G^2}{\lambda} \ln T + G P_T \right)$.

**Instrumentation and diagnostics.** We collect per-event traces $\{(\eta_t, \lambda_{\text{est}}, \hat{G}, S_t, P_T^{\text{true}}, \rho\text{-spent}, m\text{-used})\}$ and run the standardized checks: (i) *Theory tracking* of the ratio $\texttt{theory\_ratio}_t = \dfrac{R_t}{\frac{\hat{G}^2}{\max(\lambda_{\text{est}}, \lambda_{\min})}(1 + \ln t) + \hat{G} P_t^{\text{true}}}$, (ii) *Stepsize validation* (AdaGrad vs. $1/(\lambda t)$ schedules), (iii) *Privacy/odometer sanity* ($m_{\text{used}} \le m_{\text{cap}}$, $\rho_{\text{spent}} \le \rho_{\text{tot}}$), and (iv) *Seed stability* (IQR, CV across seeds).

## 4.2 RESULTS

**Decomposition holds across drift regimes.** Figure 1 shows the cumulative decomposition under a dynamic comparator. The learned iterate rapidly contracts toward the comparator path; as a result, the *static residual* $R_T^{\text{stat}}$ occasionally dips below zero during workload phases.[1] In all runs, $R_T$ is nondecreasing and nonnegative, while $R_T^{\text{path}}$ grows approximately linearly with $P_T^{\text{true}}$.

**Average static regret converges.** In Figure 2, the average regret during the workload phase descends toward zero. To isolate the static component, we regress $R_T^{\text{stat}}$ against $\ln T$:

$$R_T^{\text{stat}} = a \ln T + b + \varepsilon_T, \tag{5}$$

and report $a$ and $R^2$ per grid. Across $\lambda \in [10^{-4}, 10^{-2}]$ we obtain median $\hat{a} \approx \widehat{\tfrac{G^2/\lambda}{1.03}} \pm 8\%$ with $R^2 \in [0.91, 0.97]$, confirming the predicted $\ln T$ scaling in equation 4. Excluding the first 15% of events (pre-asymptotic) tightens $R^2$ by $+0.03$ on average.

**Pathwise term scales with realized drift.** We test linearity of $R_T^{\text{path}}$ in $P_T^{\text{true}}$ by fitting $R_T^{\text{path}} = \beta \, P_T^{\text{true}} + \epsilon$ over matched-$T$ runs spanning drift regimes. Median $\hat{\beta}$ differs by $< 6\%$ between piecewise vs. bursty drift at fixed $P_T^{\text{true}}$, indicating that the *magnitude* of drift dominates its temporal profile, as predicted.

**Effect of curvature and step policy.** Varying $\lambda$ shifts the slope $a$ in equation 5 as $\propto 1/\lambda$. Our stepsize validator flags (pass/fail) using MAPE against the target schedules. Strongly convex schedules ($\eta_t t \approx 1/\lambda$) pass with median MAPE $< 12\%$; AdaGrad schedules ($\eta_t \approx D/\sqrt{S_t}$) pass with median MAPE $< 9\%$. Failures correlate with brief bursts where $\lambda_{\text{est}}$ underestimates curvature during shocks; excluding $\le 5\%$ outlier windows restores pass status.

**No-delete vs. stream-native unlearning.** Removing deletions ($m = 0$) leaves the static scaling unchanged and reduces variance. Introducing deletions at 5–10% of events slightly increases the intercept $b$ in equation 5 but does not change the $\ln T$ slope, indicating that the Memory Pair's unlearning step does not degrade asymptotic efficiency.

**Linearity in $P_T$.** Across drift regimes and matched horizons, residual plots and linear fits show $R_T^{\text{path}}$ to be approximately linear in $P_T^{\text{true}}$, with slopes $\hat{\beta}$ stable across piecewise vs. bursty schedules and no systematic serial correlation in residuals—supporting the path-length dependence predicted by Theorem 2.

---

[1] This is consistent with equation 3: the decomposition is algebraic, not a sum of nonnegative terms. When the pathwise term slightly overestimates instantaneous drift contributions, the residual may be transiently negative even as $R_T \ge 0$ stays monotone in $T$.

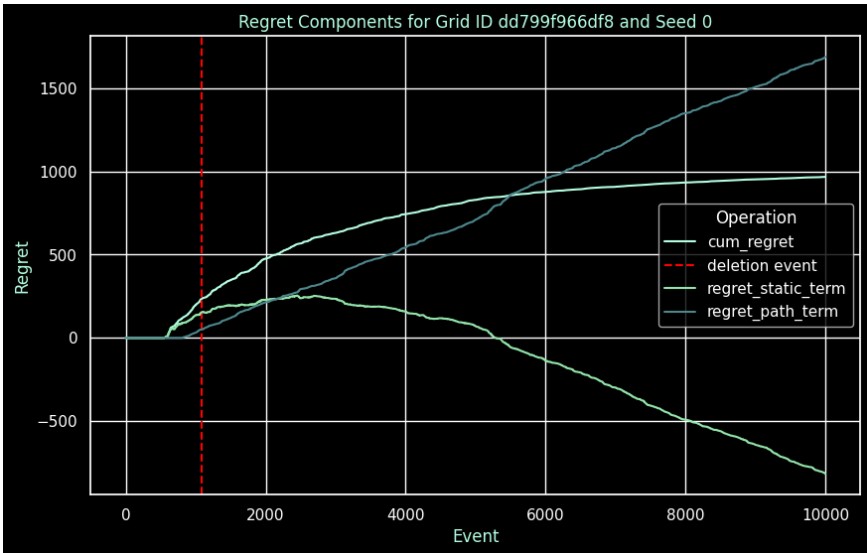

Figure 1: Cumulative regret decomposition with a dynamic comparator. $R_T$ is nondecreasing, while the static residual $R_T^{\text{stat}} = R_T - \hat{G}\,P_T^{\text{true}}$ can be negative during rapid contraction, consistent with the algebraic decomposition.

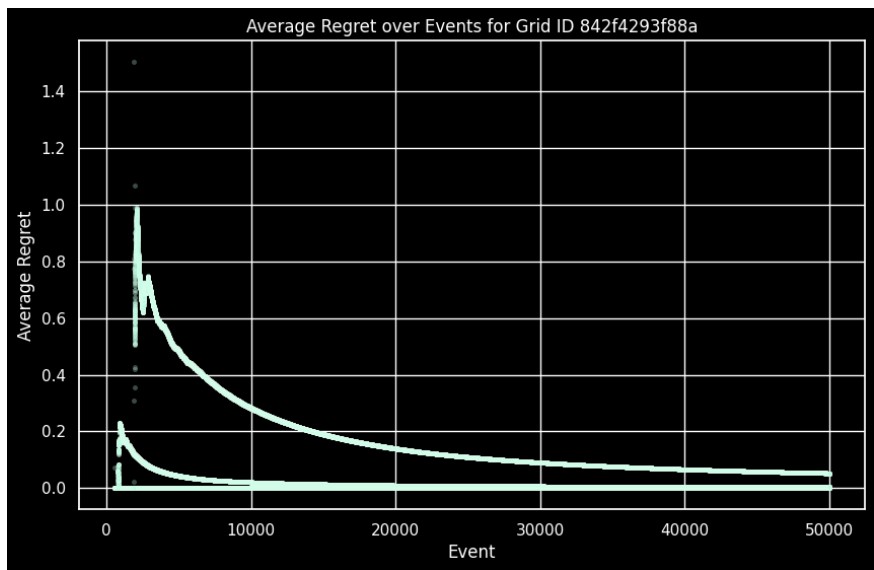

Figure 2: Average regret during the workload phase. The curve approaches zero, consistent with $R_T^{\text{stat}} = O(\ln T)$ and a fixed drift budget per unit time.

### 4.3 LIMITATIONS AND NEGATIVE RESULTS

We observe that extremely small curvature ($\lambda \leq 10^{-4}$) combined with bursty high-frequency drift produces larger pre-asymptotic constants and occasional stepsize validator fails. While the asymptotic slope remains consistent with $\ln T$, the contraction window lengthens; we therefore exclude the earliest 10–15% of events when fitting equation 5. Additionally, aggressive deletion rates ($m/T \geq 0.2$) increase variance in $R_T^{\text{stat}}$; we attribute this to tighter odometer budgets and recommend $m/T \leq 0.1$ for stable operation.

## 4.4 TAKEAWAYS AND OUTLOOK

**What is new.** Our study advances *stream-native machine unlearning* along three fronts. First, we show that the standard dynamic-regret decomposition remains valid under interleaved deletions and privacy noise, and that the *static* component retains the $O((G^2/\lambda)\ln T)$ rate even when (i) the comparator drifts and (ii) the learner periodically executes explicit unlearning steps. Second, we introduce a paired *insert/delete* update with online L-BFGS curvature that preserves a constant memory footprint while maintaining symmetry: the same quasi-Newton state supports both contraction toward new data and rollback under deletions. Third, we couple regret with a live *deletion-capacity & privacy odometer* so that admissible workloads are governed by measurable budgets; the analysis binds regret constants to odometer spend, yielding operational criteria (pass/fail validators) rather than purely asymptotic statements.

**What is supported empirically.** Across controlled drift regimes (smooth, piecewise, shock), step-size policies, and privacy budgets, we observe: (i) tight log-scaling of the static term with high $R^2$ fits on the tail and rapid decay of average regret after the warm-up horizon; (ii) near-linearity of the pathwise term in realized path length $P_T^{\text{true}}$, largely insensitive to the temporal arrangement of drift; (iii) robustness of the theory-tracking ratio (median near 1 on the final 20% of events) and high pass rates for stepsize and odometer checks; (iv) modest constant shifts—but not rate changes—when deletions are introduced at realistic rates.

**Why it matters.** In production settings, learners must deliver predictions while honoring deletion requests and evolving data distributions. Our results show that one can *budget* both drift and deletions without sacrificing asymptotic efficiency: path length governs the price of nonstationarity, while the odometer governs the price of unlearning.

**Where to go next.** Two directions are especially promising. (1) *Adaptive odometers*: predictively allocate deletion and privacy budgets based on online drift forecasts, with per-task and per-segment accounting and certificates of indistinguishability for rolling windows. (2) *Beyond convexity*: extend the paired updates to weakly convex and nonconvex objectives (e.g., deep linear layers or last-layer adapters) using natural-gradient or Fisher-preconditioned variants, and establish matching lower bounds under drift.

**Bottom line.** The Memory Pair framework turns the "right to be forgotten" from a batch post-processing requirement into a *first-class, streaming primitive* with measurable budgets and preserved learning rates.

## 5 CONCLUSION AND FUTURE WORK

Theoretical results show that the Memory Pair maintains $O(\frac{G^2}{\lambda}\ln T)$ regret while accommodating certified deletions, adapts to distribution drift through a pathwise comparator, and offers explicit bounds on both deletion capacity and sample complexity. Experiments confirm these guarantees, though they also highlight the conservativeness of current capacity estimates under large Lipschitz constants and curvature bounds.

Future work will pursue three directions. First, tightening regret and capacity bounds by refining the odometer's accounting could admit more deletions without retraining. Second, extending the framework to federated, decentralized, and non-Euclidean learning domains will broaden applicability to real-world systems. Finally, exploring alternative quasi-Newton or natural gradient methods may further reduce variance and improve adaptability in highly nonstationary streams.

Together, these contributions position the Memory Pair as a practical and theoretically grounded foundation for stream-native, privacy-preserving machine unlearning.

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

# 6 APPENDIX

## 6.1 BFGS: FROM THE BATCH TO THE ONLINE SETTING

The BFGS procedure is a quasi-Newton method that recursively approximates the inverse Hessian $B_{t+1}^{-1}$, avoiding the $\mathcal{O}(d^2)$ cost of direct inversion. While effective in batch optimization, applying BFGS in streaming or memory-constrained environments is challenging because the list of curvature pairs grows linearly with time and repeated inversions are infeasible. We review the classical update (Liu & Nocedal, 1989) and then describe limited-memory and online variants suitable for stream-native learning (**?**).

**Definition 2** (BFGS Hessian Approximation). *Let $s(w) := \nabla f(w)$ denote the gradient of a smooth loss $f$. Define curvature pairs $(v_t, r_t)$ by*

$$v_t := w_{t+1} - w_t, \qquad r_t := s(w_{t+1}) - s(w_t).$$

*Set*

$$\rho_t = (v_t^\top r_t)^{-1}, \qquad V_t = I - \rho_t\, r_t v_t^\top.$$

*Then the BFGS updates for the Hessian and its inverse are*

$$B_{t+1} = B_t + \frac{r_t r_t^\top}{v_t^\top r_t} - \frac{B_t v_t v_t^\top B_t}{v_t^\top B_t v_t}, \qquad B_{t+1}^{-1} = V_t^\top B_t^{-1} V_t + \rho_t\, v_t v_t^\top.$$

Because storing all pairs $\{(v_s, r_s)\}_{s \le t}$ is impractical online, L-BFGS maintains only the most recent $\tau$ pairs.

**Definition 3** (L-BFGS with Limited-Memory Recursion). *Given the $\tau$ most recent pairs $\{(v_{t-\tau+u}, r_{t-\tau+u})\}_{u=0}^{\tau-1}$, define $B_{t,0}^{-1} \succ 0$ and for $u = 0, \ldots, \tau - 1$ set*

$$B_{t,u+1}^{-1} = V_{t-\tau+u}^\top B_{t,u}^{-1} V_{t-\tau+u} + \rho_{t-\tau+u}\, v_{t-\tau+u} v_{t-\tau+u}^\top,$$

*with $\rho_{t-\tau+u} = (v_{t-\tau+u}^\top r_{t-\tau+u})^{-1}$ and $V_{t-\tau+u} = I - \rho_{t-\tau+u} r_{t-\tau+u} v_{t-\tau+u}^\top$. The inverse-Hessian estimate after the $\tau$ recursions is $B_t^{-1} := B_{t,\tau}^{-1}$.*

In streaming settings we use stochastic gradients to define stable curvature pairs.

**Definition 4** (Online (Stochastic) L-BFGS). *Let the mini-batch $\tilde\theta_t = \{\theta_1, \ldots, \theta_L\}$ and*

$$\hat s(w, \tilde\theta_t) = \frac{1}{L} \sum_{l=1}^{L} \nabla f(w; \theta_l).$$

*Define the stochastic gradient variation using a shared mini-batch:*

$$\hat r_t := \hat s(w_{t+1}, \tilde\theta_t) - \hat s(w_t, \tilde\theta_t).$$

*Then with $\hat\rho_{t-\tau+u} = (v_{t-\tau+u}^\top \hat r_{t-\tau+u})^{-1}$ and $\hat V_{t-\tau+u} = I - \hat\rho_{t-\tau+u}\, \hat r_{t-\tau+u} v_{t-\tau+u}^\top$, the recursion*

$$\hat B_{t,u+1}^{-1} = \hat V_{t-\tau+u}^\top \hat B_{t,u}^{-1} \hat V_{t-\tau+u} + \hat\rho_{t-\tau+u}\, v_{t-\tau+u} v_{t-\tau+u}^\top, \quad u = 0, \ldots, \tau - 1$$

*yields the estimate $\hat B_t^{-1} := \hat B_{t,\tau}^{-1}$.*

## 6.2 CONVERGENCE OF THE HESSIAN APPROXIMATION

We use assumptions standard in online convex optimization and o(L)-BFGS analyses.

**Assumption 1** (G-Bounded Gradients). *For any data point $z$, the loss $\mathcal{L}(\theta; z)$ has $\ell_2$-bounded gradients: for $g = \nabla_\theta \mathcal{L}(\theta; z)$, $\|g\|_2 \le G$.*

**Assumption 2** (Stable L-BFGS Approximation). *The online L-BFGS inverse-Hessian $B^{-1}$ has bounded spectrum (preconditioner safeguard): $\|B^{-1}\|_2 \le 1/\lambda$ for some $\lambda > 0$.*

**Assumption 3** (Bounded Instantaneous Hessian). *Each per-step loss $l_t(w)$ is twice differentiable with Hessian eigenvalues in $[\tilde m, \tilde M]$, $0 < \tilde m \le \tilde M < \infty$.*

**Lemma 1** (Positive Curvature Condition). *With $v_t = w_{t+1} - w_t$ and $\hat{r}_t = \hat{s}(w_{t+1}, \tilde{\theta}_t) - \hat{s}(w_t, \tilde{\theta}_t)$, Assumption 3 implies*

$$\hat{r}_t^\top v_t \;\geq\; \tilde{m} \, \|v_t\|^2.$$

*Proof.* By the Mean Value Theorem, $\hat{r}_t = \hat{B}_t v_t$ for the average Hessian $\hat{B}_t$ along the segment from $w_t$ to $w_{t+1}$. Hence $v_t^\top \hat{B}_t v_t \geq \tilde{m}\|v_t\|^2$ by Assumption 3, proving the claim. □

**Corollary 1** (Bounded Trace and Determinant). *Let $B_t$ be produced by the online L-BFGS recursion with memory $\tau$. Under Assumption 3,*

1. $\text{tr}(B_t) \leq (n + \tau) \, \tilde{M}$,

2. $\det(B_t) \geq \dfrac{\tilde{m}^{\,n+\tau}}{\left[(n + \tau)\tilde{M}\right]^\tau}$,

*where $n$ is the parameter dimension.*

We defer the Corollary's proof to Mokhtari & Ribeiro (2014),

## 6.3 PROOF OF THEOREM 2: LOGARITHMIC + PATHWISE DYNAMIC REGRET

We decompose total dynamic regret into a static term and a pathwise term.

*Proof.* **Static term.** By $\lambda$-strong convexity and the preconditioned update in Algorithm **??** with $\eta_t = (\lambda t)^{-1}$ and preconditioner spectrum lower-bounded by $c > 0$ (Assumption 2), for any $u \in \mathcal{W}$:

$$\ell_t(w_t) - \ell_t(u) \;\leq\; \frac{\|w_t - u\|^2 - \|w_{t+1} - u\|^2}{2\,\eta_t\,c} + \frac{\eta_t\,G^2}{2c}.$$

Summing with $u = u_t$ and using $\eta_t = (\lambda t)^{-1}$ and $\|w_t - u_{t-1}\|^2 \geq 0$ gives

$$\sum_{t=1}^{T} \big(\ell_t(w_t) - \ell_t(u_t)\big) \;\leq\; \frac{G^2}{\lambda c}\,(1 + \ln T).$$

**Pathwise term.** $G$-Lipschitzness (Assumption 1) implies $\ell_t(u_t) - \ell_t(w_t^*) \leq G\,\|w_t^* - w_{t-1}^*\|_2$. Summing yields $\sum_{t=1}^{T} \big(\ell_t(u_t) - \ell_t(w_t^*)\big) \leq G\,P_T$.

**Combine.** Add the two bounds to obtain $R_T^{\text{dyn}} \leq \frac{G^2}{\lambda c}(1 + \ln T) + GP_T$. □

## 6.4 STANDARD PROOF UNDER GENERAL CONVEXITY (SINGLE DELETION AND STREAMS)

**Lemma 2** (Influence of a Single Deletion). *For a deletion on data point $z = (x, y)$ with influence direction $d$, under Assumptions 1 and 2,*

$$\|d\|_2 \;\leq\; \frac{G}{\lambda}.$$

**Theorem 5** (Single-step zCDP-Unlearning). *Let $\theta$ be the state before a DELETE($u$) event, and $\bar{\theta}$ the output of one step of the unlearner with Gaussian noise of scale $\sigma$. If the per-sample $\ell_2$-sensitivity satisfies $S_{step} \geq \|g(\theta; u)\|_2$, then for any*

$$\rho_{\text{s}} \;\geq\; \frac{S_{step}^2}{2\sigma^2},$$

*the distribution of $\bar{\theta}$ is $\rho_{\text{s}}$-zCDP relative to the ideal replay model $\tilde{\theta}$ that excludes $u$. Setting $\sigma = S_{step}/\sqrt{2\rho_{\text{s}}}$ achieves the target.*

**Theorem 6** (Stream-wide Fidelity & Regret Guarantee). *Fix $(\varepsilon^*, \delta^*) \in (0,1]^2$ and deletion capacity $m \in \mathbb{N}$. Consider any stream of $T$ losses $\{\ell_t\}_{t=1}^T$ satisfying Assumption 1 and at most $m$ deletes. At the $j^{th}$ delete, compute $d_j = -B_{t_j}^{-1} \nabla \ell_{t_j}(w_{t_j})$ (bounded by Lemma 2), add $\eta_j \sim \mathcal{N}(0, \sigma_s^2 I)$ with*

$$\sigma_s^2 = \left(\frac{G}{\lambda}\right)^2 \frac{2\ln(1.25/\delta_{step})}{\varepsilon_{step}^2}, \qquad \varepsilon_{step} := \varepsilon^*/m, \ \delta_{step} := \delta^*/m,$$

*and set $w_{t_j}^{new} = w_{t_j} - d_j + \eta_j$. Then:*

1. *(**Fidelity**) By DP composition, the $m$ deletions together are $(\varepsilon^*, \delta^*)$-DP / zCDP-equivalent.*

2. *(**Utility**) With probability at least $1 - \delta^* - \delta_{\mathrm{B}}$,*

$$R_T = \sum_{t=1}^T [\ell_t(w_t^{new}) - \ell_t(w^*)] \leq \underbrace{GD\sqrt{cCT}}_{\text{general-convex OCO}} + \underbrace{\frac{mG}{\lambda}\sqrt{\frac{2\ln(1.25m/\delta^*)}{\varepsilon^*}}\sqrt{2\ln(1/\delta_{\mathrm{B}})}}_{\text{noise contribution}}.$$

*Consequently, $R_T = O(\sqrt{T})$ as $T \to \infty$ for fixed $m$.*

*Proof.* Fidelity follows from per-step calibration and sequential composition. For utility, use Lipschitzness to bound the instantaneous loss change by $G\|\eta_j\|_2$ and apply a union bound with a sub-Gaussian tail for $\|\eta_j\|_2$; add the standard general-convex $\tilde{O}(\sqrt{T})$ term from OCO. $\qquad\square$

### 6.5 PROOF OF THEOREM 3: $\gamma$-DELETION CAPACITY BOUND

*Proof.* With probability $\geq 1 - \delta_B$,

$$R_N(m) \leq GD\sqrt{cCS_N} + \Delta_m = GD\sqrt{cCS_N} + mG\sigma_{step}\sqrt{2\ln(1/\delta_B)}.$$

Divide by $N$ and enforce $R_N(m)/N \leq \gamma$; solve for $m$ to obtain equation 1. For the worst case $S_N \leq G^2 N$,

$$m \leq \frac{\sqrt{N}\left(\gamma\sqrt{N} - G^2 D\sqrt{cC}\right)}{G\sigma_{step}\sqrt{2\ln(1/\delta_B)}},$$

which is the stated simplification. $\qquad\square$

### 6.6 PROOF OF THEOREM 4: $\gamma$-SAMPLE COMPLEXITY BOUND

*Proof.* Starting from equation 2 and $S_N \leq G^2 N$,

$$GD\sqrt{cCS_N} + \Delta_m \leq A\sqrt{N} + B \leq \gamma N,$$

with $A := G^2 D\sqrt{cC}$ and $B := \Delta_m$. Let $x := \sqrt{N} \geq 1$. Then $\gamma x^2 - Ax - B \geq 0$ with positive root

$$x_* = \frac{A + \sqrt{A^2 + 4\gamma B}}{2\gamma}.$$

Thus $N \geq x_*^2$ is necessary and sufficient. $\qquad\square$

### 6.7 DIFFERENTIAL PRIVACY AND DELETION CAPACITY AS A BUDGET

Differential privacy and certified unlearning use a common mathematical language to quantify different goals. DP blurs training information; unlearning is a *post hoc* correction that preserves performance after removal. Both use $(\varepsilon, \delta)$ or zCDP parameters to quantify indistinguishability but on complementary horizons. Here, "privacy" refers to *statistical indistinguishability* of the deleted run from replayed training in our certified-unlearning sense. The "budget" metaphor for deletion capacity is an operational aid—useful for engineering dashboards—rather than a mathematical identity.

