# OpenReview forum: "Stream Native Machine Unlearning"
_ICLR.cc/2026/Conference — Submitted to ICLR 2026_

### Official Review · Reviewer_q2tK · 2025-10-27

**Soundness:** 1
**Presentation:** 1
**Contribution:** 2
**Rating:** 2
**Confidence:** 4

**Summary:**

This paper proposes a theoretical framework for machine unlearning in streaming environments, where learning and unlearning occur continuously rather than in static, i.i.d. training settings. The authors introduce the concept of a Memory Pair, coupling an online learner and unlearner that share curvature information through an online L-BFGS mechanism. Their framework aims to provide logarithmic regret bounds under strong convexity and certified deletion guarantees based on differential privacy.
Key theoretical results include bounds on static and dynamic regret, deletion capacity, and sample complexity, and the paper also presents a simple experimental validation using synthetic non-stationary streams to demonstrate convergence of the regret decomposition.

**Strengths:**

The paper tackles an important and timely question: how to formalize continual unlearning in realistic streaming environments where models must learn and unlearn simultaneously. The authors make a commendable effort to develop a theoretical foundation for this setting, combining tools from online convex optimization, regret analysis, and differential privacy.
The introduction of memory pairs provides an appealing abstraction that unifies learning and unlearning as symmetric operations, and the incorporation of a deletion-capacity odometer to balance fidelity, privacy, and utility is conceptually interesting. The theoretical analysis attempts to make the notion of continual unlearning tractable and quantifiable, and the paper’s use of regret and DP arguments to jointly analyze learning performance and deletion fidelity is novel and potentially useful.

**Weaknesses:**

- Overall clarity and readiness of presentation - The manuscript appears somewhat rushed and unpolished, and not at the level of completeness typically expected for an ICLR submission. There are numerous broken or missing references, and the exposition is often unclear. Several notations (for example, $R_N$) are used well before being defined, and many definitions (such as that of Deletion Capacity) are vague or ambiguous, leaving it uncertain whether they hold only under specific assumptions or for a finite number of deletions. The writing would benefit from significant editing to improve readability, logical flow, and precision.

- Unclear algorithmic framework - The algorithmic formulation in Section 2 raises conceptual questions. The “unlearn step” is defined as taking a single noisy gradient ascent step with respect to the deleted sample, but in continual learning, one might have multiple prior learning steps for the same data. It is unclear how a single reverse update could fully undo such cumulative influence. The authors should clarify what is actually being unlearned and how this corresponds to retraining without the sample, or whether it merely approximates it.

- Weak and limited experimental evaluation - The empirical validation is minimal and confined to synthetic data with controlled drift, which does not convincingly demonstrate the practical usefulness of the framework. The evaluation does not test the unlearning capability in realistic or challenging scenarios, for example, situations involving backdoor removal, label corruption, or real continual-learning benchmarks (see, e.g., https://arxiv.org/abs/2405.18040). Such experiments would be essential to substantiate the claims about the framework’s applicability and robustness.

**Questions:**

see above

---

### Official Review · Reviewer_qJfJ · 2025-10-28

**Soundness:** 2
**Presentation:** 1
**Contribution:** 2
**Rating:** 2
**Confidence:** 3

**Summary:**

This paper proposes Memory Pair, a stream-native learning–unlearning framework using online L-BFGS. It achieves logarithmic regret under strong convexity, certified unlearning via zCDP, and a deletion-capacity odometer, enabling constant-memory, on-device operation that sustains accuracy amid interleaved deletions and drift, reducing costly retraining.

**Strengths:**

1. The unlearning in a stream training environment deserves to be researched
2. This paper proposes a theoretical guarantee for the logarithmic regret and proves that the proposed method is certified unlearning
3. This paper conducts a numerical empirical validation to show that the method maintains accuracy under mixed insert and delete workloads

**Weaknesses:**

1. This paper relies on strong assumptions (λ-strong convexity, G-Lipschitz losses, bounded L-BFGS spectra), limiting applicability to non-convex deep models.
2. This paper does not compare any other static unlearning approaches with similar assumptions.
3. The experiment setup is not clear, especially for the data generation and model construction.
4. In Figure 2, the curve approaches zero, which cannot prove that the regret bound is linear to $O(ln(T))$. It can only prove it is sublinear to $T$
5. Apart from the synthetic data, more real-world datasets are encouraged to be used to show how the proposed method works when the assumptions get weaker.
6. This paper does not discuss the obstacles of applying online learning techniques to the unlearning problem, which can reduce the novelty of this paper.
7. Many typos exist in this paper, for example, ?? on line 242.
8. This paper does not provide proof for Theorem 1.

**Questions:**

Please refer to the weaknesses

---

### Official Review · Reviewer_ec1a · 2025-11-01

**Soundness:** 3
**Presentation:** 3
**Contribution:** 2
**Rating:** 4
**Confidence:** 4

**Summary:**

The paper introduces Memory Pair, the first stream-native algorithm for machine unlearning—a setting where models must simultaneously learn from and forget data in a non-stationary streaming environment. Unlike prior work that assumes static, batch-style training, this framework explicitly treats learning and unlearning as dual online processes sharing curvature information through an online L-BFGS quasi-Newton state.

Key contributions:
	1.	Unified learning–unlearning pair: a symmetric update rule handling both insert and delete operations via shared quasi-Newton memory.
	2.	Deletion-capacity odometer: an adaptive accounting mechanism linking deletion frequency, regret bounds, and differential-privacy (zCDP) budgets.
	3.	Constant-memory implementation: enabling on-device or edge deployment while preserving theoretical fidelity and privacy guarantees.
	4.	Theoretical guarantees: proofs of logarithmic regret under λ-strong convexity, dynamic-regret decomposition for drift, and explicit bounds on deletion capacity m and sample complexity N.

**Strengths:**

Originality
	•	The concept of a stream-native unlearning framework is highly original. Prior certified-unlearning methods (Bourtoule et al., 2019; Neel et al., 2020; Sekhari et al., 2021) operate in static regimes, whereas this paper generalizes to continuous streams with theoretical regret guarantees.
	•	Introducing paired insert/delete symmetry and a privacy-budget odometer represents a creative synthesis of online convex optimization and differential-privacy accounting—a novel and conceptually elegant bridge between two previously disjoint areas.
Quality
	•	The theoretical development is rigorous. The paper derives regret bounds under strong convexity (Theorems 1–4) and shows precise scaling of the deletion capacity m and sample complexity N.
	•	Experiments are carefully instrumented: the authors track theory ratios, step-size validation, privacy spend, and seed stability—rare in this domain. The decomposition between static and pathwise regret is empirically verified across multiple drift regimes.
Clarity
	•	Despite the heavy mathematical content, the exposition is generally clear and disciplined.
	•	Definitions (e.g., Memory Pair, deletion-capacity odometer) are formalized early and consistently used.
	•	Algorithm 1 concisely expresses the symmetry between learning and unlearning; figures (e.g., regret decomposition) nicely reinforce the theoretical story.
Significance
	•	Practically, the work advances privacy-preserving continual learning, relevant to federated, on-device, and IoT systems where storage and privacy constraints preclude retraining.
	•	Theoretically, it introduces a unifying language for regret and certified-unlearning guarantees, which could influence future research on adaptive privacy budgets and dynamic-data compliance.

**Weaknesses:**

1.	Experimental realism:
	•	The evaluation uses synthetic drifting streams rather than real-world datasets (e.g., sensor or user-event logs). Demonstrating the method on practical streaming benchmarks would bolster its empirical credibility.
	2.	Comparative baseline gap:
	•	No direct experimental comparison with recent online or incremental unlearning algorithms (e.g., Qiao et al., 2024; Waerebeke et al., 2025). Quantitative baselines would help contextualize the claimed efficiency and regret improvements.
	3.	Computational cost reporting:
	•	The constant-memory claim is convincing theoretically, but runtime or wall-clock comparisons (vs. e.g., SGD or Online Newton Step) are missing. Explicit profiling would substantiate the “on-device feasibility” argument.
	4.	Strong-convexity assumption:
	•	The entire analysis hinges on λ-strong convexity; extension to non-convex neural objectives remains an open question. This limitation should be emphasized more clearly in the main text.
	5.	Terminological density:
	•	Some sections (e.g., Section 2.3 and 3) use very compact notation that might deter non-theorist readers. Adding intuition or visual diagrams for the odometer and regret decomposition could enhance accessibility.

**Questions:**

1.	Adaptive odometer scheduling:
	•	Could deletion budgets ρ be allocated adaptively based on observed drift or gradient variance rather than uniformly?
	2.	Extension to non-convex settings:
	•	The authors mention “future work” on extending to weakly convex or Fisher-preconditioned variants. Are there preliminary results or intuitions on whether curvature sharing remains stable for deep models?
	3.	Comparative evaluation:
	•	How does Memory Pair perform against Hessian-free unlearning (Qiao et al., 2024) in terms of fidelity and runtime on similar streaming workloads?
	4.	Memory and compute profiling:
	•	What is the actual per-event computational overhead of maintaining online L-BFGS curvature pairs (τ = memory window)? How does this scale with parameter dimension d?
	5.	Broader deployment implications:
	•	Could the framework integrate with federated settings where deletes arrive asynchronously from multiple clients? How would the odometer aggregate cross-device budgets?

---

### Official Review · Reviewer_GQyz · 2025-11-01

**Soundness:** 3
**Presentation:** 3
**Contribution:** 3
**Rating:** 4
**Confidence:** 3

**Summary:**

The paper proposes an online unlearning framework that utilizes a paired learning and unlearning mechanism that shares the same model state and curvature memory. Using an online L-BFGS approximation, the algorithm maintains a constant memory footprint while performing incremental parameter updates for both data insertion and deletion events. The deletion operation is modeled as a reversed optimization step with added differential privacy noise, and a “deletion capacity odometer” is proposed to account for the cumulative privacy budget and regulate how many deletions the system can perform safely.

**Strengths:**

1. Long term online unlearning is a solid topic.

2. The idea of coupling learning and unlearning through shared curvature memory is conceptually appealing. It provides a unified framework where insertions and deletions are treated symmetrically under a single optimization process.

3. The paper establishes regret bounds and demonstrates that interleaved learning and unlearning do not cause model instability. The use of online convex optimization theory to prove bounded regret adds mathematical rigor to the framework.

**Weaknesses:**

1. The experiments only report regret convergence and model stability under deletions. There are no evaluations demonstrating that deleted samples are truly forgotten, such as changes in model predictions, reduced membership inference attack success, or behavioral indistinguishability from retrained models. Without such experiments, the claim of achieving certified unlearning remains unsubstantiated.

2. All theoretical guarantees rely on convexity and Lipschitz continuity. Lipschitz continuity is more then acceptable but convexity on the other hand is not so much. The authors briefly mention extending to non-convex objectives as future work but do not provide any indication that the current approach can generalize beyond simple convex optimization tasks.

3. The method depends on a sliding curvature window that stores only a limited number of recent updates. If a deletion request arrives long after a data point was inserted, the relevant gradient and curvature information have already been overwritten. In such cases, the algorithm has no basis for computing the correct reverse update, and the so-called unlearning operation becomes a rough approximation at best. This makes the approach unsuitable for long-term or asynchronous data streams.

4. The very mechanism that enables fast reversibility, namely the retention of recent curvature pairs, introduces potential privacy risks. The cached gradients can be accessed or analyzed by an insider or attacker to infer properties of the original training data. As a result, the system may still hold sensitive information even after a deletion operation has been performed. This outcome runs contrary to the intended goal of privacy-preserving unlearning.

5. The algorithm operates online, but the learning and unlearning processes are still serialized and sequential. The model cannot simultaneously learn new data and forget old data in a truly continuous fashion. In this sense, the “stream-native” label is largely cosmetic: it presents a traditional optimization loop with alternating update signs rather than a genuinely adaptive or concurrent streaming system.

**Questions:**

Please refer to the weaknesses.

---

### Meta-Review · Area_Chair_2ZyC · 2026-01-04

**Summary:**

1. No evaluation demonstrating that the forget set is completely unlearned.
2. The algorithm operates online, but the learning and unlearning processes are still serialized and sequential. Thus, the whole system cannot be "stream native".
3. No direct experimental comparison with recent online or incremental unlearning algorithms.
4. The non-convex extension requires substantive development, not just a mention in future work.

**Reviewer Concerns:**

The authors did not provide a rebuttal.

**Reviewer Scores:**

The authors did not provide a rebuttal.

---

### Decision · Program_Chairs · 2026-01-26

Reject